# The Improvement of Dispersion Stability and Bioaccessibility of Calcium Carbonate by Solid/Oil/Water (S/O/W) Emulsion

**DOI:** 10.3390/foods11244044

**Published:** 2022-12-14

**Authors:** Jie Zhang, Gongwei Li, Yanping Cao, Duoxia Xu

**Affiliations:** Beijing Advanced Innovation Center for Food Nutrition and Human Health (BTBU), School of Food and Health, Beijing Engineering and Technology Research Center of Food Additives, Beijing Higher Institution Engineering Research Center of Food Additives and Ingredients, Beijing Key Laboratory of Flavor Chemistry, Beijing Laboratory for Food Quality and Safety, Beijing Technology & Business University, Beijing 100048, China

**Keywords:** calcium carbonate, solid/oil/water (S/O/W) emulsions, physicochemical stability, microstructure, rheological behavior, bioaccessibility

## Abstract

Solid/oil/water (S/O/W) emulsion loaded with calcium carbonate (CaCO_3_) was constructed to raise the dispersion stability and bioaccessibility. In the presence or absence of sodium caseinate (NaCas), the particle size, Zeta-potential, physical stability, and apparent viscosity of stabilized S/O/W emulsions with different gelatin (GEL) concentrations (0.1~8.0 wt%) were compared. Combined with a confocal laser scanning microscope (CLSM), cryoscanning electron microscope (Cryo-SEM), and interfacial adsorption characteristics, the stabilization mechanism was analyzed. The bioavailability of CaCO_3_ was investigated in a simulated gastrointestinal tract (GIT) model. The S/O/W-emulsion droplets prepared by the NaCas–GEL composite have a smaller particle size, higher Zeta-potential, larger apparent viscosity, and better physical stability compared with GEL as a single emulsifier. CLSM results confirmed that CaCO_3_ powder was encapsulated in emulsion droplets. The Cryo-SEM results and interfacial adsorption characteristics analysis indicated that the NaCas–GEL binary composite could effectively reduce the interfacial tension, and the droplets form a denser three-dimensional network space structure with a shell–core structure which enhanced the stability of the system. GIT studies showed that the droplets presented higher CaCO_3_ bioaccessibility than the CaCO_3_ powder. This study enriched the theory of the S/O/W transfer system and provided theoretical support for the development of CaCO_3_ application in liquid food.

## 1. Introduction

Emulsion is regarded as a simple and effective way of embedding and delivering nutrients, which has attracted the interest of researchers in the food industry. The emulsion delivery system can effectively improve the stability and digestibility of nutrients [1,2,3,4,5]. Water-soluble functional factors are usually directly added to food in the form of a solution and are absorbed by the human body during consumption, with good bioavailability [6]. Fat-soluble functional ingredients are limited by their poor solubility. Emulsions provide a new idea for their application in food systems. However, mineral elements such as calcium, iron, and zinc have weak solubility and low digestibility and absorptivity, seriously limiting their application. S/O/W-emulsion droplets are commonly known as food emulsions containing the solid phase. The S/O/W (solid/oil/water) emulsions are constructed as a carrier for the solid-phase nutrient [7,8,9]. In S/O/W technology, the S phase is added to the O phase, originating a suspension-like liquid, and then the W phase is added. With the water-soluble emulsifier, shearing, and homogenization, S/O/W-emulsion droplets are prepared. As a carrier of food nutrients, the S/O/W-emulsion droplets have a low cost and simplicity. At present, the research using S/O/W emulsions is mainly focused on enzyme preparations and probiotic carriers [10,11,12,13,14], while there are few studies on the bioavailability and controlled-release characteristics [15,16].

Calcium carbonate (CaCO_3_) is the ingredient choice for calcium supplements with a high calcium content and a low price [17]. However, at the same time, calcium carbonate has disadvantages, such as poor solubility, easy precipitation, and gastric irritation [18]. Studies have suggested embedded calcium-rich milk mineral salts with lecithin so that microencapsulated calcium will not be hydrolyzed by stomach acid in stomach digestion [19], which could improve the bioavailability of calcium [20]. Nowadays, milk calcium fortified by microencapsulation technology has been added to many milk powders formulas [21]. In the field of food, milk powders are mainly fortified with calcium carbonate, and the calcium supplement products on the market are mostly calcium carbonate. The main problem of food products added with insoluble calcium salts is that they tend to precipitate out and produce precipitation during storage. The researches on carriers for the delivery of calcium carbonate mainly include nanocalcium carbonate–sodium alginate gelatin balls [22], nanocalcium carbonate Pickering emulsion [23], etc. Studies have been effectively conducted to improve both the stability and the absorption rate of calcium. Nevertheless, there are no studies reported on traditional CaCO_3_ powder in liquid foods.

Protein is one of the main raw materials used to prepare S/O/W emulsions and plays an emulsifying and stabilizing role. Proteins such as whey protein, sodium caseinate, and gelatin are used to stabilize emulsions [24,25,26]. The stabilization mechanism may be due to the increased viscoelastic barrier formed by the interaction of protein and emulsifier, which prevents the binding of the emulsion droplets and the loss of coated compounds in the inner emulsion droplets. Gelatin (GEL) is a natural food thickener with excellent charge properties, and gelation characteristics. GEL contains hydrophilic and lipophilic groups, has surface activity, and is used as an emulsifier. GEL could create a steady oil–water interface and is effective in avoiding droplet accumulation [27]. It was found that increasing the concentration of GEL could increase the apparent viscosity and modulus of the Pickering emulsion and form a denser network structure to improve the emulsion stability [28]. Internal aqueous-phase gelation using whey isolate protein or gelatin can improve the stability and encapsulation efficiency of W/O/W double emulsions [29,30]. However, when GEL is used alone as an emulsifier, relatively large droplets will be formed, resulting in poor system stability [31,32], and its application in food processing is limited. Sodium caseinate (NaCas) can reduce the interfacial tension and form a relatively thick interface layer by adsorbing on the oil–water interface and preventing the aggregation and flocculation of oil droplets through the electrostatic exclusion and steric hindrance effect [33,34]. Composite biofilms or gel microspheres prepared using NaCas–GEL have a wide range of applications in food packaging and biopharmaceuticals [35,36,37].

Few studies on the application of traditional CaCO_3_ powder in liquid food have been reported. In this study, a S/O/W emulsion is used to deliver CaCO_3_ to improve the dispersion stability and bioaccessibility. The S/O/W emulsion was stabilized by GEL and NaCas–GEL as the W phase to load CaCO_3_. The droplet size, microstructure, rheological behavior, and interface behavior were systematically evaluated. This study aimed to investigate the stabilization mechanism of S/O/W emulsions stabilized by GEL and NaCas and to provide a theoretical basis for the delivery of CaCO_3_ by the S/O/W emulsion and enrich the theoretical support for the development of a new nutrient-delivery system.

## 2. Materials and methods

### 2.1. Materials

CaCO_3_ was acquired from Shuang Teng Industrial Co., Ltd. (Zhengzhou, China). Sodium caseinate (Lot #C10185129) was obtained from Macklin Biochemical Co., Ltd. (Shanghai, China). Gelatin was bought from Sinopharm Chemical Reagents Co., Ltd. (Beijing, China). Soybean Oil was provided by COFCO Co., Ltd. (Tianjin, China). Nile Blue A and Nile Red were procured from Sigma-Aldrich Co. (St. Louis, MO, USA).

### 2.2. S/O/W Emulsion Preparation

The solid phase (S) was CaCO_3_ powder and the oil phase (O) was soybean oil, with the weighed ratio of 1:10, and stirred at 700 rpm for 1 h. The mixtures were stirred for 3 min at 15,000 rpm with a high-speed mixer (ULTRA TURRAX T25 Digital, IKA, Staufen, Germany) to form the S/O phase suspension. The outer water-phase (W) solutions were prepared by dissolving NaCas (10.0 wt%) and GEL (2.0 wt%, 8.0 wt%) in PBS buffer (1.0 mM, pH 7.0). Different concentrations of the GEL and NaCas solution were accurately weighed (1:1, *w/w*) and stirred magnetically for 2 h to obtain the W phase. The W phase was mixed with the S/O phase suspension (5%:95%, *w*/*w*), and the mixtures were then stirred at 15,000 rpm for 5 min with a high-speed disperser to obtain the S/O/W emulsions.

### 2.3. Particle Size, Zeta-Potential, and Physical Stability Measurements

Particle size measurements were performed using a laser diffractometer (S3500 Microtrac Inc., Largo, FL, USA) with the particle refractive index of 1.51 and the dispersive medium refractive index of 1.33.

Zeta-potential was performed using a Zetasizer Nano-ZS90 (Malvern Instruments, Worcestershire, UK). The electrophoretic mobility was measured in 11 consecutive readings after the sample was equilibrated for approximately 120 s.

Physical stability measurements were performed using a LUMiSizer (LUM GmbH, Berlin, Germany). The injection volume was 0.4 mL, the rotation speed was 2000 rpm, the time interval was 10 s, the spectral line was 200, and the test temperature was kept at 25 °C.

### 2.4. Viscosity Measurements

A HAAKE rheometer (MARS IQ Air, HAAKE, Vreden, Germany) was used to measure the apparent viscosity of the samples. When adding the samples, pay attention to keeping the sample uniform to prevent bubbles. The parameters were as follows: plate rotor CC25 DIN, the used gap was 4 mm, the shear rate was 2~200 s^−1^, the test temperature was kept at 25 °C.

### 2.5. Microstructure

#### 2.5.1. Confocal Laser Scanning Microscopy (CLSM)

The CLSM (FV3000, Olympus, Tokyo, Japan) was used to observe the microstructure of the samples. Before sample preparation, soybean oil was dyed with Nile red (488 nm laser excitation source) and the protein was dyed with Nile blue A (635 nm laser excitation source), then stored overnight at low temperature and protected from light. The samples were magnified using a 10× eyepiece and a 60× objective lens (oil immersion), and the microscopic images were obtained and processed by software Olympus.

#### 2.5.2. Cryo-Scanning Electron Microscopy (Cryo-SEM)

Cryo-SEM (Helios NanoLab G3 UC, FEI, Hillsboro, OR, USA) was employed in the examination of the interfacial structure. Samples were first prefrozen in liquid nitrogen, then transferred to freezer preparation chamber (PP3010T, Quorum Technologies, Lewes, UK). Freeze-fracturing, vacuum sublimation, then sputter-coated with platinum were completed under the vacuum condition. Finally, the samples were transferred to scanning electron microscope (SEM) for observation.

### 2.6. Interfacial Behavior of Interaction between NaCas and GEL

#### 2.6.1. Oil-phase Purification

The edible soybean oil was used as the O phase. Soybean oil contains a small amount of surface-active ingredients and some other impurities, which generally have to be purified first to prevent affecting the results. The purification operation was as follows: the Floresil molecular sieve adsorbent and the soybean oil was mixed at a ratio of 3% (*w*/*w*) [38] and magnetically stirred for 2 h, then centrifuged (10,000 rpm, 25 °C, 20 min) to remove the precipitation. New adsorbent was added and the above operation repeated. After a repeated operation, the interfacial tension of phosphate buffer on purified corn oil did not change significantly within 30 min. The density of the purified oil phase was 0.8744 g/cm^3^, and the interfacial tension for phosphate buffer was 27.3 ± 0.5 mN/m.

#### 2.6.2. Interfacial Tension

Interfacial tension meter (Biolin Scientific, Goteborg, Germany, Attention Theta Flax) was used to observe the change of the interfacial pressure (π) with time (*t*) and the adsorption kinetics of the NaCas and NaCas–GEL solutions at the oil–water interface. Firstly, the samples were added to a microsampler with an outer diameter of 2.08 mm, removing the air bubbles to exclude them from affecting the measurement results. Secondly, soybean oil was poured into a rectangular quartz glass jar and a sampling needle was inserted, whereby the autosampling system pushed 12 μL samples to the tip of the needle to form a complete droplet. The camera system immediately and continuously captured the change of the droplet profile and calculated the interfacial tension of the sample at different time points in real-time, and the whole process lasted for 180 min. The experimental temperature was maintained at 25 °C, the interference of external vibration was avoided as much as possible during the test, and all samples were repeated three times.

The dynamic interfacial tension is expressed by the surface pressure (π) as a function of adsorption time:π = *σ*_0_ − *σ_t_*
where *σ*_0_ is the surface tension of buffer solution and *σ_t_* is the surface tension of samples.

#### 2.6.3. Rheological Properties of Interfacial Membrane Expansion

The shaking drop technique was used to study the changes in interfacial expansion modulus (E), interfacial expansion elasticity (E_d_), interfacial expansion viscosity (E_v_), and phase angle (θ) of the NaCas and NaCas–GEL complexes adsorbed on the oil–water interface with the adsorption time.

After droplet formation, sinusoidal oscillations were generated by an oscillation generator, and images of the droplet shape were acquired periodically by camera image contour and analyzed by the system software for image analysis. When the droplet oscillated sinusoidally with the appropriate frequency and amplitude, the surface area of the droplet changed slightly (expansion strain), resulting in a change in the surface tension (expansion stress). In this experiment, the frequency was set at 0.1Hz and the amplitude was set at 10%.

### 2.7. Simulated Gastrointestinal Tract (GIT) Model

Gastrointestinal digestion and bioavailability of S/O/W emulsions were studied using a GIT model. An automatic digestive system (GI20, Australia NI Instrument, Ryde, Australia), which includes mouth, stomach, and small intestine, was employed [39].

The initial emulsions (5 mL) were added into GIT system with 4 mL SSF (pH = 7.0), 0.75 mL α-amylase (1000 U/mL), and 0.25 mL ultrapure water. The pH was adjusted to 6.8, the mixture was kept at 37 °C, and then it was shaken at 100 rpm for 10 min.

The oral-digested samples were mixed with 8 mL SGF and 0.667 mL pepsin (5000 U/mL). The pH was adjusted to 3.0, the mixture was kept at 37 °C, and then it was shaken at 100 rpm for 2 h.

The stomach-digested samples were mixed with 8.0 mL SIF, 5.0 mL pancreatic enzyme (800 U/mL), 3.0 mL bile salt solution (200 mg/mL), and 4 mL deionized water, respectively. The pH was adjusted to 7.0, the mixture was kept at 37 °C, and then it was shaken at 100 rpm for 2 h.

### 2.8. Calcium Bioaccessibility

The bioaccessibility of CaCO_3_ was measured using an inductively coupled plasma optical emission spectrometer (ICP-OES) (Optima 8000, PerkinElmer, Waltham, MA, USA) after in vitro digestion. The enzymes in the intestine digestion products were inactivated in a 100 °C water bath for 15 min and then centrifuged at 10,000 rpm for 40 min to collect the micelle phase. Secondly, a calibration blank was made with ultrapure water, followed by a standard linear value with a standard calcium solution having a series of concentration gradients (*R*^2^ > 0.999). The micelle phase was diluted 100 times with ultrapure water and calcium concentration was measured at a wavelength of 422.67 nm. The following equation was used to calculate calcium bioaccessibility [40]:$$\mathrm{Calcium}\;\mathrm{biozccessibility}\;(\%)\;=\;\frac{\mathrm{soluble}\;\mathrm{calcium}\;(\mathrm{mg}/L)}{\mathrm{total}\;\mathrm{calcium}\;\mathrm{in}\;\mathrm{matrix}\;(\mathrm{mg}/L)}\;\times \;100\%$$

### 2.9. Statistical Analysis

Each experiment was performed at least in triplicate. The results were presented as means ± standard deviations. The plots and data analysis were drawn by Origin 8.5 software (Origin Lab., Northampton, MA, USA) and the SPSS 17.0 statistical analysis system (SPSS Inc., Chicago, IL, USA).

## 3. Results and Discussion

### 3.1. Physical Stability

The LUMiSizer Stability analyzer can quickly determine the stability of emulsions by accelerating stratification, quantitative precipitation, and suspension [41]. In the original transmittance curve, the red curve at the bottom is the first spectral line for sample stability analysis, and the green one at the top is the last spectral line for sample scanning. The shift of the spectral line from left to right indicates precipitation, and the shift from right to left indicates an emulsion creaming phenomenon.

The light transmittance of the samples with different GEL concentrations encapsulating CaCO_3_ are shown in Figure 1a. As the GEL concentration increased, the change of transmittance gradually decreased. When the GEL concentration was low, the light transmittance was higher, indicating that the sample was unstable during the centrifugal process, and the migration of the dispersed phase occurred. At this time, the GEL was insufficient to cover the surface of the S/O/W-emulsion droplets. With the GEL concentration increased, the light transmittance of the sample decreased, and the S/O/W emulsion maintained good physical stability. When the NaCas was added to the W phase, the transmission curves of the S/O/W emulsions stabilized by GEL and NaCas were shown as seen in Figure 1b. The addition of NaCas decreased the transmission of the sample, indicating that NaCas was advantageous to improving the stability of the S/O/W emulsions.

The instability index of the S/O/W emulsions with different GEL concentrations encapsulating CaCO_3_ are shown in Figure 1c. As the GEL concentration increased, the instability index decreased. When the GEL concentration was low, the instability index was large. The lowest value (about 0.01) was obtained when the GEL concentration was 8 wt%, because the emulsification effect of the sample was enhanced at a higher GEL concentration, the viscosity increased, and the stability of the system increased. When NaCas was added into the W phase, the instability index decreased, indicating that NaCas was advantageous to improving the S/O/W emulsion stability.

### 3.2. Particle Size

The particle-size distribution of the samples prepared by the different GEL concentrations showed double peaks (Figure 2a), indicating that samples prepared by the GEL has poor stability with large droplets [42,43]. The low GEL concentration was not sufficient to cover the surface of the S/O/W-emulsion droplets and promoted the aggregation of droplets. When the GEL concentration increased, the particle size distribution shifted left and became narrower, indicating that the particle size decreased generally, and the stability of the system was improved. When NaCas was added into the W phase, the bimodal phenomenon disappeared. The overall particle size distribution shifted to the left and became narrower (Figure 2b), indicating that NaCas was beneficial to improving the stability of the S/O/W emulsion. The average particle size of the S/O/W emulsion was larger at a low GEL concentration, and the smallest average particle size (3.60 ± 0.71 μm) was obtained when the GEL concentration was 8 wt% (Figure 2c), indicating the best emulsification effect and the most stable emulsion state. When NaCas was added to the W phase, the average particle size could only be reduced and the S/O/W emulsion stability improved, which was the result of NaCas–GEL binary complex.

### 3.3. Zeta-Potential

The zeta-potential is an electrostatic repulsion to characterize the stability of the S/O/W-emulsion droplets. In general, the higher the absolute value (positive or negative) of the zeta-potential, the more stable the system (dissolution or dispersion can resist aggregation). On the contrary, the lower the zeta-potential (positive or negative), the more unstable the system is and the more inclined to condensation (the attractive force exceeds the repulsive force), and the dispersion is destroyed. The effects of the different GEL concentrations on the zeta-potential of the S/O/W-emulsion microspheres is shown in Figure 3. The zeta-potential of the samples stabilized by different concentrations of the GEL were all negative. The pH of the S/O/W-emulsion liquid system was 7.0, and the pI of the GEL was 7.5~9. At a low GEL concentration (0.1 wt%), the zeta-potential value was only −7.92 ± 0.21 mV. With the GEL concentration increased, zeta-potential became −14.33 ± 1.44 mV (8 wt%), and the stability of the system was enhanced. When NaCas was added to the W phase (pI was 4.6), the zeta-potential increased, and the stability was improved.

### 3.4. Viscosity Analysis

Apparent viscosity is closely related to the stability of the emulsion system. The rheological property of emulsion is closely related to its internal system and reflects its microstructure. The effects of different GEL concentrations on the apparent viscosity of the droplets are shown in Figure 4. The apparent viscosity decreased with increasing shear rate and then gradually became stable, indicating that the system has pseudoplastic fluid properties. The network structure was damaged when the shear rate increased, which led to the orderly arrangement according to the flow direction, reducing the flow resistivity and the apparent viscosity [44]. The apparent viscosity increased as the GEL concentration increased, and the GEL molecules played a thickening role and constituted to the network structure, which led to the increase of system stability. When NaCas was added to the W phase, the GEL permeated into the NaCas network structure, resulting in the increased structural strength of the spatial network and the improved the S/O/W emulsion stability.

### 3.5. Microstructure Analysis

The S/O/W emulsions prepared by the XG and NaCas–GEL binary composite were all milky white under visual observation and showed no signs of separation or foaming, and it is difficult to distinguish the difference. Therefore, the microstructure of the samples prepared by the NaCas and NaCas–GEL should be further observed in order to study its embedding effect and interface adsorption more directly.

#### 3.5.1. CLSM

The CLSM images characterized the formation of the samples (Figure 5a). The O phase displayed green, the W phase displayed red, and the black part of the inner oil phase was CaCO_3_. Without NaCas, the droplets prepared by GEL (2 wt%) had a wide size range, low zeta-potential, and the system was unstable. When the GEL concentration increased (8 wt%), the particle size distribution was more uniform and the particles were smaller, which were consistent with the results in Figure 2. When NaCas was present in the W phase, the size of the droplets prepared by the GEL (2 wt%) decreased, the interaction between GEL and NaCas was lessened, the system was relatively unstable, and the encapsulation effect of the CaCO_3_ was poor. At a high GEL concentration (8 wt%), the GEL and NaCas interaction increased, the particle size distribution was more uniform and the particles were smaller, the stability was improved, and the encapsulation effect of the CaCO_3_ was enhanced. Magnifying the results of the external aqueous protein staining showed that the presence of NaCas enhanced the thickness of the adsorption layer, which was advantageous to the formation of microspheres and the stability of the system.

#### 3.5.2. Cryo-SEM

Cryo-SEM can be used for visualizing the microsphere structure at the emulsion interfaces [42]. Figure 5b shows the cross-sectional structure of the samples. The S/O/W emulsions prepared with the GEL (2 wt%) and without the NaCas showed the lamellar structure of the W phase, where particles of a different size of the O phase were embedded. When the GEL concentration was increased (8 wt%), the samples formed a compact honeycomb three-dimensional spatial network structure, and the droplets were uniformly distributed. The steric hindrance effect generated by the network structure can prevent the flocculation of the droplets and improve the stability of the system. When NaCas was present in the W phase, the S/O/W-emulsion microsphere system prepared at a low concentration (2 wt%) formed a honeycomb three-dimensional network structure, and the three-dimensional network structure was denser at a high concentration of GEL (8 wt%), which was advantageous to improving the stability of the system. At the same time, the cross-section of microspheres showed that the presence of NaCas enhanced the thickness of the adsorption layer and forming a core–shell structure, which led to the improvement of the physical stability of the system [45], and the results were consistent with those in Figure 5a.

### 3.6. Interface Adsorption Characteristics

#### 3.6.1. Interfacial Tension

Protein has a direct impact on emulsification performance and therefore plays an important role in oil–water emulsions [46]. The primary adsorption of proteins at the interface is the most significant stage in emulsion formation [47]. In order to better understand whether the formation of the NaCas–GEL mixture in aqueous solution leads to the difference in adsorption behavior of interfacial protein components, the dynamic adsorption performance of NaCas, GEL, and NaCas–GEL was studied, and the π-t^1/2^ relation curve is shown in Figure 6a. The π value gradually increased with the extension of the adsorption time, indicating that the complex formed by NaCas, GEL, and NaCas–GEL gradually adsorbed onto the oil–water interface [48,49]. The result showed that the presence of GEL had a significant influence on the interfacial pressure generated by NaCas adsorption at the oil–water interface. Compared with NaCas, the interfacial pressure of the NaCas–GEL mixture was higher.

Generally, the adsorption of proteins on the oil–water interface consists of three main stages, namely, diffusion, penetration, and molecular rearrangement [50]. The dynamic adsorption, infiltration, and rearrangement of proteins at the oil–water interface have an important impact on their emulsification properties and emulsification stability [51,52]. With the extension of adsorption time, the π-t^1/2^ curve deviates completely from the straight line when the surface pressure is high. In other words, diffusion no longer controls the adsorption kinetics at high interfacial protein concentrations. This is because the energy barrier for diffusion of protein molecules increases with the increase of π value. The adsorption rate is determined by the ability of protein molecules to gain space at the interface, allowing them to unfold and rearrange.

The equation can be used to analyze the penetration and rearrangement adsorbed on the interface [53]:$$\ln[\left(\pi_{10,800}\;-\;\pi_{t}\right)/\left(\pi_{10,800}\;-\;\pi_{0}\right)]\;=\;-k_{i}t$$
where π_10,800_, π*_t_*, and π_0_ represent the interfacial pressure when the adsorption time is 10,800 s, *t* s, and 0 s, respectively, and *k_i_* is the first-order rate constant. In general, the fitted curve produces two or more linear regions. The slope of the first linear region corresponds to the rate at which protein molecules fold at the interface (*k*_P_), and the slope of the second linear region corresponds to the rate constant at which protein molecules rearrange at the interface (*k*_R_).

The folding and rearrangement rates of protein molecules in NaCas, GEL, and the NaCas–GEL complex systems at the oil–water interface were shown in Figure 6b. The results showed that, in the simple NaCas system, the initial bulk phase value had no significant effect on the osmotic folding rate of the protein molecules at the interface; that is, the ability of protein molecules to enter the interface phase and fold on the interface was not affected by the value. The π value increases the fastest in the initial stage of adsorption, and with the extension of the adsorption time, the increase amplitude of the π value decreases gradually. The absolute values of the proteins at *k*_R_ are all higher than that of *k*_P_, which indicates that the structural rearrangement of adsorbed particles at the interface plays a dominant role in the formation of the interfacial film. The interaction between the NaCas and GEL may occur through the crosslinking physical adsorption of hydrophobic groups, so the shielding of hydrophobic groups may be the cause of the reduced penetration and rearrangement rate.

#### 3.6.2. Rheological Properties of Interface Expansion

Figure 6c showed the evolution of E with π in the surface layer for the adsorption of NaCas and NaCas–XG mixed films. The E−π curve and its slope can give information about the adsorption capacity of particles on the interface and the interaction between molecules [54]. The slope of the E−π curve is about 1, indicating that the protein is ideally adsorbed on the interface at this time. The increase of E is only related to the amount of protein adsorbed on the interface, and there is no obvious interaction between protein particles. A slope value for the E−π curve greater than 1 indicates that the adsorption of components on the interface is nonideal and there is strong intermolecular interaction. The slope value of the E−π curve is less than 1, which indicates that the adsorption of components on the interface is nonideal and there is weak intermolecular interaction. Results showed that the E increased with the interfacial pressure, and this dependence reflects the existence and the increase of interactions between film-forming components. The longer the adsorption time, the higher the E, which was in agreement with the theory of Lucassen–Reynders. The slope values for the E−π curves reported in Figure 6c are all higher than 1.0, which indicates a strong interaction between the interfacial membrane components compared to the ideal adsorption behavior. The slopes of the E−π curves of the NaCas–GEL mixtures were all higher than those of GEL, indicating that the presence of NaCas had a significant effect on the molecular structure and/or condensation (stacking) of NaCas adsorbed at the oil–water interface. NaCas increased the E value of the GEL interfacial absorption layer, which indicated that the interaction between the NaCas and GEL can improve the mechanical strength of the oil–water interfacial absorption layer, which has good interfacial stability for stabilizing the S/O/W emulsion.

### 3.7. Calcium Bioaccessibility

The calcium bioaccessibility of the S/O/W-emulsions droplets was studied, which was compared with CaCO_3_ powder [55]. Previous studies showed that the samples prepared by GEL (8 wt%) and NaCas–GEL (8 wt%) had high apparent viscosity and good physical stability. In vitro digestion experiments simulated the calcium bioaccessibility after digestion, emulsions were prepared by GEL and NaCas–GEL, and CaCO_3_ powder, and the results were shown in Figure 7. The S/O/W emulsions prepared by the GEL and NaCas–GEL had a higher bioaccessibility than the CaCO_3_ powder. This is because during the gastric digestion phase, CaCO_3_ was converted to Ca^2+^ by gastric acid, and the proteins in the W phase were digested by gastric fluid to produce peptide molecules [56]. The specific peptide combines with Ca^2+^ to form a soluble complex, which prevents Ca^2+^ from forming insoluble precipitates, and then the Ca^2+^ can reach the absorption site, promoting calcium absorption in the intestine. The Ca^2+^ produced by pure CaCO_3_ powder forms more insoluble calcium precipitates. The results show that the S/O/W emulsion can be used as a good delivery carrier for CaCO_3_ and improve the bioavailability of calcium.

## 4. Conclusions

In this study, S/O/W emulsions to deliver CaCO_3_ using GEL and NaCas–GEL complexes were fabricated. Compared with GEL as a single emulsifier, the samples prepared by the NaCas–GEL binary composite had smaller particle sizes, higher zeta-potential, higher apparent viscosity, and better physical stability. The CLSM results confirmed that CaCO_3_ was encapsulated to form S/O/W-emulsion microspheres. The Cryo-SEM and interfacial adsorption characteristics analysis results showed that the NaCas–GEL binary composite could effectively reduce the interfacial tension, and the S/O/W-emulsion droplets formed a denser three-dimensional network space structure with a shell–core structure which enhanced the stability of the system. In addition, the S/O/W emulsions improved the bioaccessibility of CaCO_3_ in the simulated GIT model. This design of S/O/W emulsions can overcome the disadvantage of poor solubility of CaCO_3_ and be applied to the delivery of liquid-food minerals.

## Figures and Tables

**Figure 1 foods-11-04044-f001:**
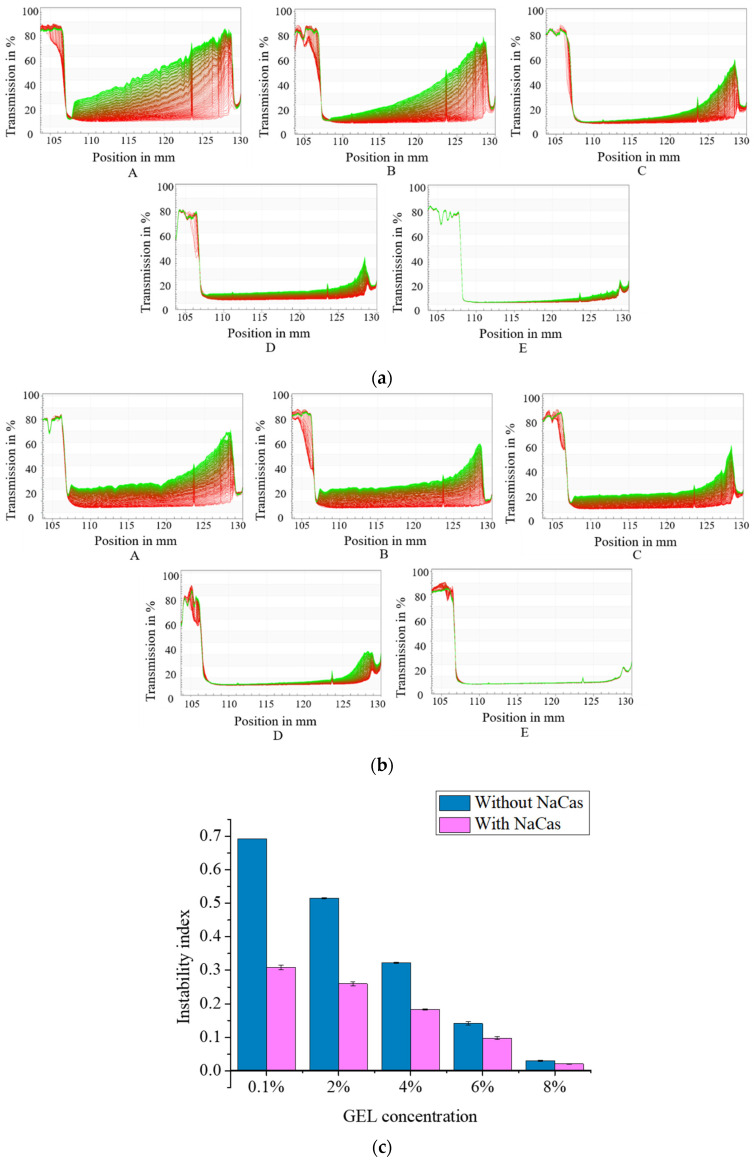
LUMiSizer analysis of CaCO_3_ S/O/W emulsions prepared by different GEL concentrations (0.1, 2, 4, 6, 8 wt%): (**a**) without NaCas; (**b**) with NaCas; (**c**) instability index. The red and green sections are all the transmittance curves.

**Figure 2 foods-11-04044-f002:**
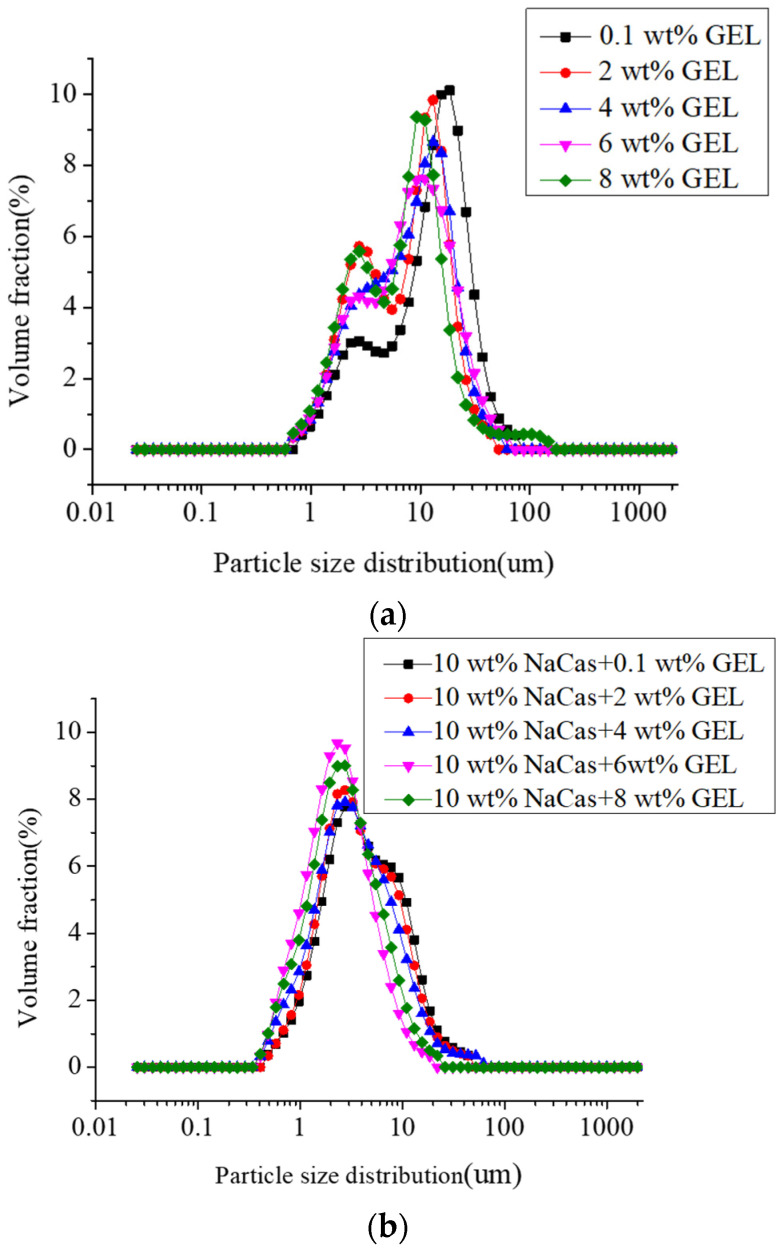
Particle size analysis of CaCO_3_ S/O/W emulsions prepared by different GEL concentrations (0.1, 2, 4, 6, 8 wt%). Particle size distribution—(**a**) without NaCas; (**b**) with NaCas; (**c**) mean particle size.

**Figure 3 foods-11-04044-f003:**
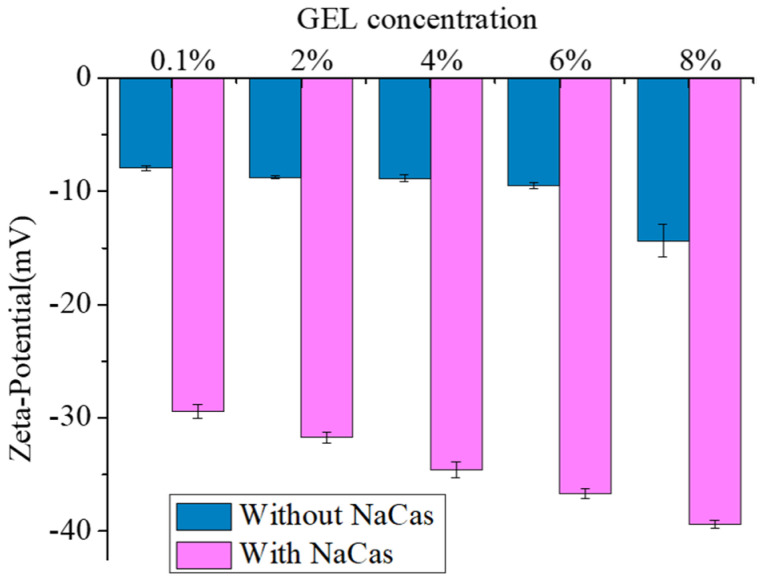
Zeta−potential analysis of CaCO_3_ S/O/W emulsions prepared by different GEL concentrations (0.1, 2, 4, 6, 8 wt%).

**Figure 4 foods-11-04044-f004:**
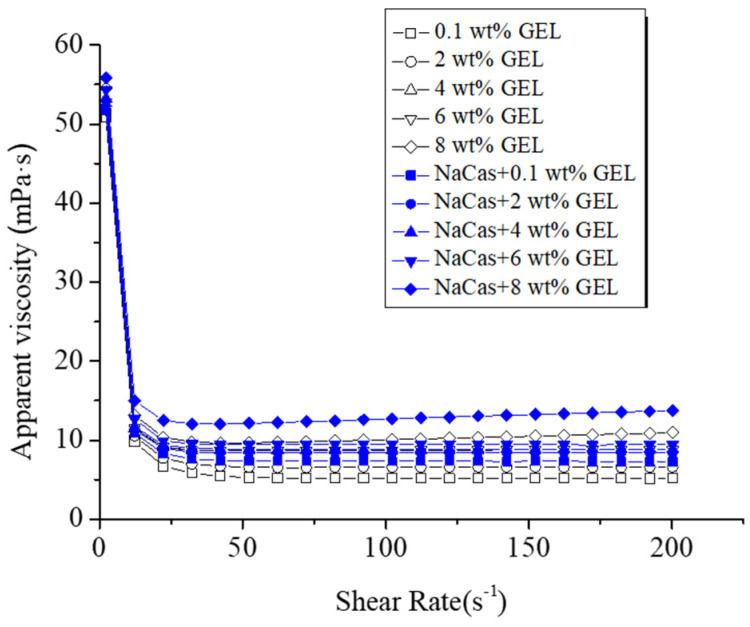
Effect of the different GEL concentrations (0.1, 2, 4, 6, 8 wt%) on the apparent viscosity of CaCO_3_ S/O/W emulsions.

**Figure 5 foods-11-04044-f005:**
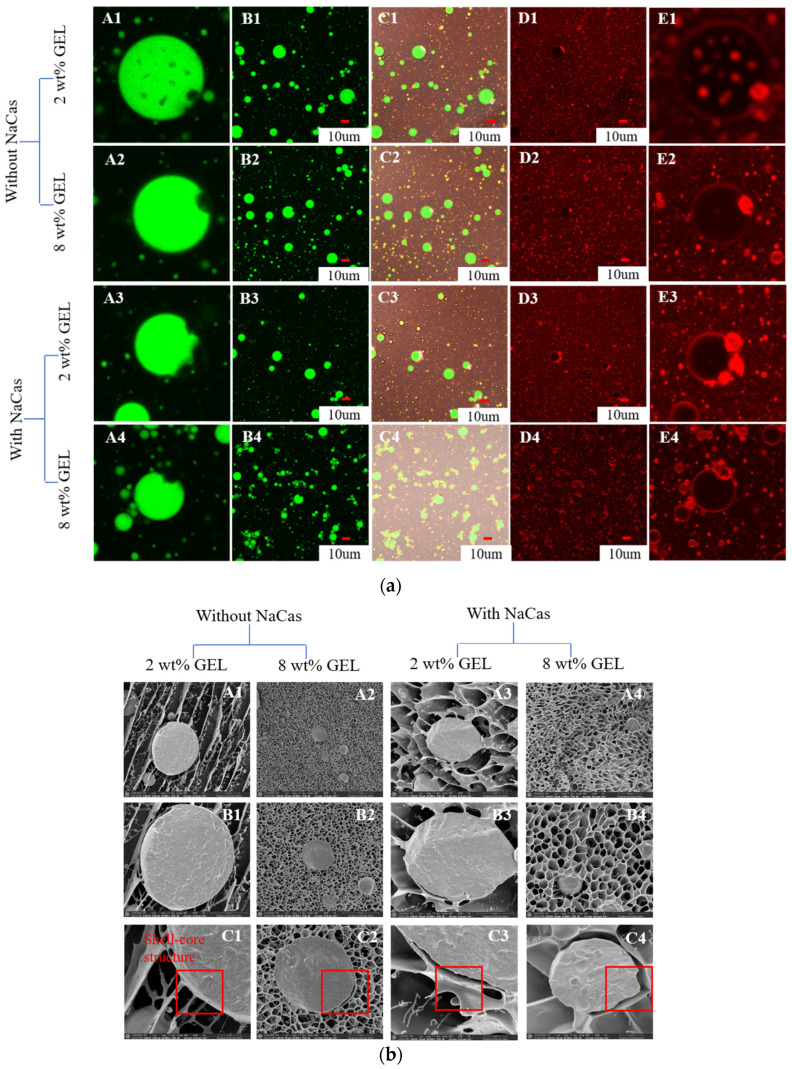
(**a**) CLSM images of the CaCO_3_ S/O/W emulsions (A/B Oil phase was stained with Nile red, excitation at 488 nm; D/E, protein phase was stained with Nile blue, excitation at 635 nm (ii); C, combined image; columns A and E state that are details of the images of columns B and D, respectively). Scale bar: 10 µm. (1, 2: 2, and 8 wt% GEL, without NaCas; 3,4: 2, and 8 wt% GEL, with NaCas). (**b**) Cryo-SEM images of the CaCO_3_ S/O/W emulsions (A, 5000×; B, 20,000×; C, 50,000×); (1, 2: 2, and 8 wt% GEL, without NaCas; 3,4: 2, and 8 wt% GEL, with NaCas).

**Figure 6 foods-11-04044-f006:**
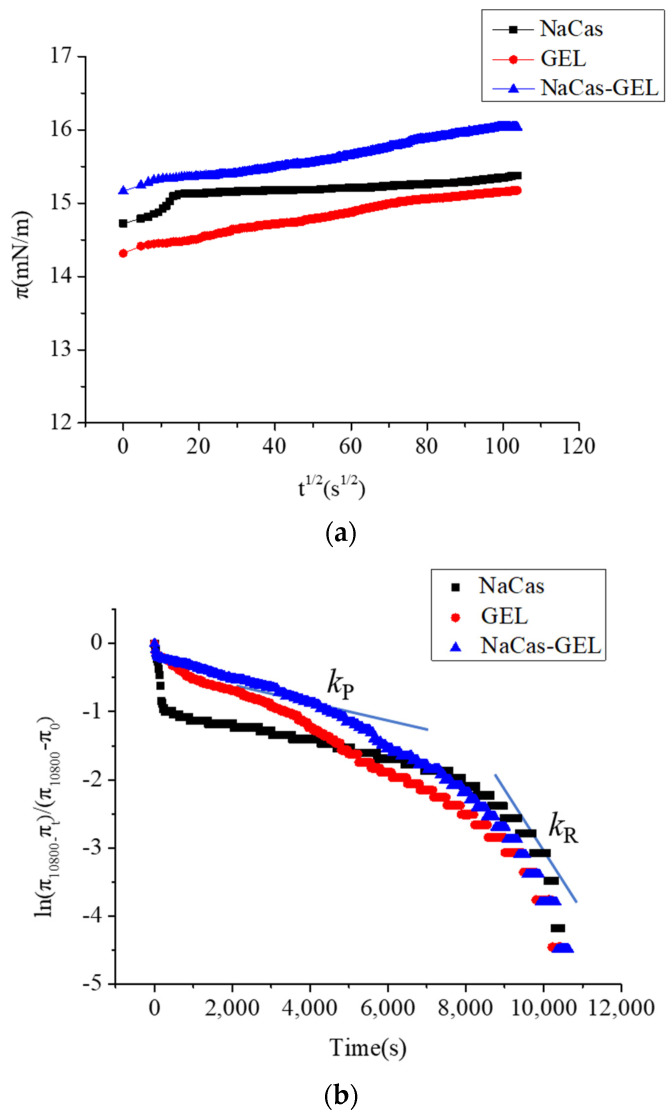
Interface adsorption characteristic at the oil–water interface: (**a**) Interfacial pressure (π) at the oil−water interface as a function of time (t1/2); (**b**) Molecular permeation and structural rearrangement at the oil–water interface; (**c**) Surface−dilatational modulus (E) as a function of surface pressure (π).

**Figure 7 foods-11-04044-f007:**
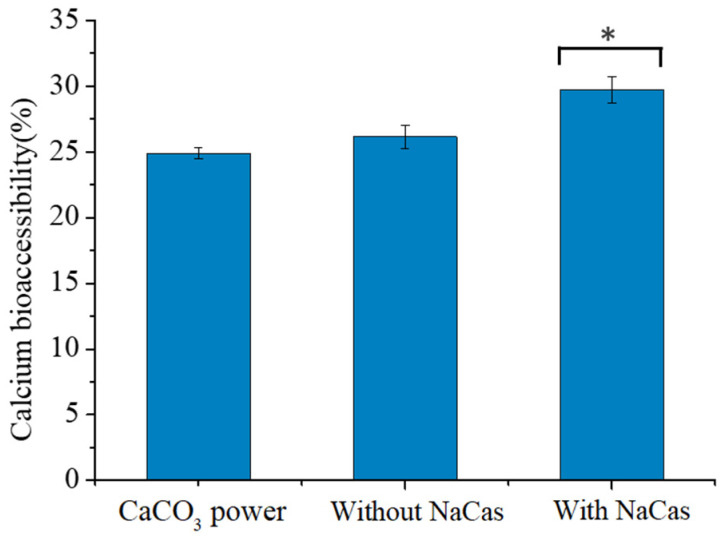
Soluble calcium concentration in the micelle phase and calcium bioaccessibility of pure CaCO_3_ powder and CaCO_3_ S/O/W emulsion. * *p* < 0.05.

## Data Availability

The data presented in this study are available on request from the corresponding author.

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
