# Peer review of "The Improvement of Dispersion Stability and Bioaccessibility of Calcium Carbonate by Solid/Oil/Water (S/O/W) Emulsion"

_foods, 2022, doi:10.3390/foods11244044_

Round 1

Reviewer 1 Report

The manuscript entitled “The improvement of dispersive stability and bio accessibility of calcium carbonate by S/O/W emulsion microspheres” reports the preparation of O/W emulsions, stabilized by gelatin and, in some cases, also by sodium caseinate, containing calcium carbonate suspended in the oil phase. The prepared systems were thoroughly characterized employing a wide palette of techniques.

I find only a weakness in the work, namely the theoretical part. I think that the used method is not suited for the study of the interaction between two macromolecules. It is more commonly used to gain information on the binding interaction between a small molecule ligand and a protein  It appears that the docking analysis suggests that the interaction takes place between two small moieties of the two proteins. Actually in Figure 7 fragments with a very low number of atoms are represented. They look different from common amino acids, in figure 7 a there is a molecule that bears a SO3- group. 

Thus, I suggest to completely omit Section 3.6.

The writing is not very fluent and in some instances inaccurate. Hereafter I am reporting some suggestions:

-the term microspheres to address the emulsion droplets looks misleading also because of the presence of solid calcium carbonate particles, therefore I suggest to use the word droplet throughout the text

-I would find clearer the alternative title “The improvement of dispersion stability and bioaccessibility of calcium carbonate by S/O/W emulsion”

-line 39 better “ingredients” than “factors”

-line 46 “originating” rather than “to diffuse into”

-line 51 delete”the construction”

-line 56 “precipitation” instead of “to precipitate”

-line 57 add “suggested” before “embedded”

-line 58 better “hydrolyzed” than “decomposed”

-line 60 move “formulas” after “milk powder”

-line 64 add “the” between “that” and “insoluble”

-line 65 better “The researches on carriers for the delivery of…”

-line 75 “charge” instead of “charged”

-line 89 delete interactions

-line 118 state that the value of refractive index of 1.51 is referred to the particles

-line 132 HAAKE, like above

-line 134 S: small letter

-the used gap should be reported

-line 143 software Olympus?

-line 146 better “employed in the examination of the …”

-line 149 delete “to sublimation” and “and then”

-line 150 “then” instead of “the samples were”

-line 154 “amount” instead of “number”

-line 159 better: “New adsorbent was added and the above operation repeated.”

-Delete Section 2.7 (see above)

-line 202-205 better “The simulated GIT model was used to investigate the gastrointestinal digestion and the bioavailability of CaCO3 of the S/O/W emulsion. An automatic digestive system (GI20, Australia NI  Instrument, Australian), which includes mouth, stomach, and small intestine phases, was employed.”

-line 208 delete “respectively” and add “kept” between “was” and “at 37 °C”

-the same at le 211 and 214-215

-line 218: 3 in CaCO3 as subscript and the same thought the text

-line 226: “calcium” in place of “lutein”

-line 241: delete “spectral  lines”

-line 242: “are shown” better  than “were shown” 

-line 242-243: better “As the GEL concentration increased, the change of transmittance ….”

-line 245: ”the migration of the dispersed phase occurred” rather than “a milk chromatography phenomenon appeared”

-line 251: “transmission” instead of “transmissibility”

-line 251: “advantageous” instead of “conductive” and also at lines 262, 354, 380

-line 258: better “large. The lowest” than “large, and the lowest”

-line 259: delete “which was”

-line 284: replace “, and” by “.”

-Figures 2 and 3 are exchanged

-line 295: it is enough to state “it is a measure of electrostatic repulsion”

-line 333: better “network” than “lattice”

-Figure 4: delete the scale on the left, it is already reported on the right side

-Figure 5 (a), columns A and E: state that are details of the images of columns B and D, respectively

-line 368: “can be used” instead of “ could available”

-line 369: delete “adsorption condition”

-lines 370-372: better “The results showed that the S/O/W prepared with GEL (2 wt%) and without NaCas showed lamellar structure of the W phase where particles of different size of the O phase were embedded”

-line 379: “and” instead of “while”

-line 381: delete “adsorption”

-line 383: better “forming a core-shell structure, which leads to the improvement”

-line 448: better “A slope value for the E-π curve greater than one indicates”, and analogously at line 456

-line 455: better “The slope values for the E-π curves, reported in Figure 6c, are all “

-line 485: add the relevant reference, 54?

-line 491: “2+” as upperscript

-line 509: “denser” instead of “more dense”

-line 510: delete: certain thickness”

-line 513: better just “emulsions” than “ emulsion microspheres”

Author Response

注释:

题为“通过 S/O/W 乳液微球改善碳酸钙的分散稳定性和生物可及性”的手稿报道了 O/W 乳液的制备,该乳液由明胶稳定,在某些情况下,还由含有碳酸钙的酪蛋白酸钠稳定悬浮在油相中。使用广泛的技术对准备好的系统进行了彻底的表征。

I find only a weakness in the work, namely the theoretical part. I think that the used method is not suited for the study of the interaction between two macromolecules. It is more commonly used to gain information on the binding interaction between a small molecule ligand and a protein. It appears that the docking analysis suggests that the interaction takes place between two small moieties of the two proteins. Actually in Figure 7 fragments with a very low number of atoms are represented. They look different from common amino acids, in figure 7 a there is a molecule that bears a SO3- group. Thus, I suggest to completely omit Section 3.6.

The writing is not very fluent and in some instances inaccurate. Hereafter I am reporting some suggestions:

Response:

-Thank you for your great suggestion on improving the accessibility of our manuscript, we have omitted Section 3.6.

Comments 1:

-the term microspheres to address the emulsion droplets looks misleading also because of the presence of solid calcium carbonate particles, therefore I suggest to use the word droplet throughout the text.

Response 1:

-Thank you for your great suggestion on improving the accessibility of our manuscript, we have used the word “droplet” instead of “microspheres.”

Comment 2:

-I would find clearer the alternative title “The improvement of dispersion stability and bioaccessibility of calcium carbonate by S/O/W emulsion.”

Response 2:

-Thank you for your great suggestion on improving the accessibility of our manuscript, we have changed the title to “The improvement of dispersion stability and bioaccessibility of calcium carbonate by solid/oil/water (S/O/W) emulsion.”

 Comment 3:

-line 39 better “ingredients” than “factors”

-Thank you for your great suggestion on improving the accessibility of our manuscript, we have changed “factors” to “ingredients.”

-line 46 “originating” rather than “to diffuse into”

-Thank you for your great suggestion on improving the accessibility of our manuscript, we have changed “to diffuse into” to “originating.”

-line 51 delete “the construction”

-Thank you for your great suggestion on improving the accessibility of our manuscript, we have deleted “the construction.”

-line 56 “precipitation” instead of “to precipitate”

-Thank you for your great suggestion on improving the accessibility of our manuscript, we have changed “to precipitate” to “precipitation.”

-line 57 add “suggested” before “embedded”

-Thank you for your great suggestion on improving the accessibility of our manuscript, we have added “suggested” before “embedded”

-line 58 better “hydrolyzed” than “decomposed”

-Thank you for your great suggestion on improving the accessibility of our manuscript, we have changed “decomposed” to “hydrolyzed.”

-line 60 move “formulas” after “milk powder”

-Thank you for your great suggestion on improving the accessibility of our manuscript, we have moved “formulas” after “milk powder”

-line 64 add “the” between “that” and “insoluble”

-Thank you for your great suggestion on improving the accessibility of our manuscript, we have added “the” between “that” and “insoluble”

-line 65 better “The researches on carriers for the delivery of…”

-Thank you for your great suggestion on improving the accessibility of our manuscript, we have redescribed the sentence: The researches on carriers for the delivery of calcium carbonate mainly include nano-calcium carbonate-sodium alginate gelatin balls [21], nano-calcium carbonate Pickering emulsion [22], etc.

-line 75 “charge” instead of “charged”

-Thank you for your great suggestion on improving the accessibility of our manuscript, we have changed “charged” to “charge”

-line 89 delete interactions

-Thank you for your great suggestion on improving the accessibility of our manuscript, we have deleted “interactions”

-line 118 state that the value of refractive index of 1.51 is referred to the particles

-Thank you for your great suggestion on improving the accessibility of our manuscript, we have redescribed the sentence: Particle size measurements were performed using a laser diffractometer (S3500 Microtrac Inc, Largo USA) with the particle refractive index of 1.51 and the dispersive medium refractive index of 1.33.

-line 132 HAAKE, like above

-Thank you for your great suggestion on improving the accessibility of our manuscript, we have corrected it.

-line 134 S: small letter

-Thank you for your great suggestion on improving the accessibility of our manuscript, we have corrected it.

-the used gap should be reported

-Thank you for your great suggestion on improving the accessibility of our manuscript, we have added it.

-line 143 software Olympus?

-Thank you for your great suggestion on improving the accessibility of our manuscript, we have added it.

-line 146 better “employed in the examination of the …”

-Thank you for your great suggestion on improving the accessibility of our manuscript, we have redescribed the sentence: Cryo-SEM (Helios NanoLab G3 UC, FEI, USA) was employed in the examination of the interfacial structure of the samples.

-line 149 delete “to sublimation” and “and then”

-Thank you for your great suggestion on improving the accessibility of our manuscript, we have deleted “to sublimation” and “and then”

-line 150 “then” instead of “the samples were”

-Thank you for your great suggestion on improving the accessibility of our manuscript, we have changed “the samples were” to “then”

-line 154 “amount” instead of “number”

-Thank you for your great suggestion on improving the accessibility of our manuscript, we have changed “number” to “amount”

-line 159 better: “New adsorbent was added and the above operation repeated.”

-Thank you for your great suggestion on improving the accessibility of our manuscript, we have changed to “New adsorbent was added and the above operation repeated”

-Delete Section 2.7 (see above)

-Thank you for your great suggestion on improving the accessibility of our manuscript, we have deleted Section 2.7.

-line 202-205 better “The simulated GIT model was used to investigate the gastrointestinal digestion and the bioavailability of CaCO3 of the S/O/W emulsion. An automatic digestive system (GI20, Australia NI Instrument, Australian), which includes mouth, stomach, and small intestine phases, was employed.”

-Thank you for your great suggestion on improving the accessibility of our manuscript, we have redescribed the sentence: Gastrointestinal digestion and bioavailability of S/O/W emulsions were studied using a GIT model. An automatic digestive system (GI20, Australia NI Instrument, Australian), which includes mouth, stomach, and small intestine, was employed.

-line 208 delete “respectively” and add “kept” between “was” and “at 37 °C”

-Thank you for your great suggestion on improving the accessibility of our manuscript, we have

deleted “respectively” and added “kept” between “was” and “at 37 °C”

-the same at le 211 and 214-215

-Thank you for your great suggestion on improving the accessibility of our manuscript, we have

deleted “respectively” and added “kept” between “was” and “at 37 °C”

-line 218: 3 in CaCO3 as subscript and the same thought the text

-Thank you for your great suggestion on improving the accessibility of our manuscript, we have corrected it.

-line 226: “calcium” in place of “lutein”

-Thank you for your great suggestion on improving the accessibility of our manuscript, we have corrected it.

-line 241: delete “spectral lines”

-Thank you for your great suggestion on improving the accessibility of our manuscript, we have deleted it.

-line 242: “are shown” better than “were shown” 

-Thank you for your great suggestion on improving the accessibility of our manuscript, we have changed “were shown” to “are shown”

-line 242-243: better “As the GEL concentration increased, the change of transmittance ….”

-Thank you for your great suggestion on improving the accessibility of our manuscript, we have redescribed the sentence: As the GEL concentration increased, the change of transmittance gradually decreased.

-line 245: “the migration of the dispersed phase occurred” rather than “a milk chromatography phenomenon appeared”

-Thank you for your great suggestion on improving the accessibility of our manuscript, we have changed “a milk chromatography phenomenon appeared” to “the migration of the dispersed phase occurred”

-line 251: “transmission” instead of “transmissibility”

-Thank you for your great suggestion on improving the accessibility of our manuscript, we have changed “transmissibility” to “transmission”

-line 251: “advantageous” instead of “conductive” and also at lines 262, 354, 380

-Thank you for your great suggestion on improving the accessibility of our manuscript, we have changed “l conductive” to “advantageous”

-line 258: better “large. The lowest” than “large, and the lowest”

-Thank you for your great suggestion on improving the accessibility of our manuscript, we have changed “large, and the lowest” to “large. The lowest”

-line 259: delete “which was”

-Thank you for your great suggestion on improving the accessibility of our manuscript, we have deleted it.

-line 284: replace “, and” by “.”

-Thank you for your great suggestion on improving the accessibility of our manuscript, we have changed it.

-Figures 2 and 3 are exchanged

-Thank you for your great suggestion on improving the accessibility of our manuscript, we have changed it.

-line 295: it is enough to state “it is a measure of electrostatic repulsion”

-Thank you for your great suggestion on improving the accessibility of our manuscript, we have redescribed the sentence: Zeta-potential is an electrostatic repulsion to characterize the stability of the S/O/W emulsion droplets.

-line 333: better “network” than “lattice”

-Thank you for your great suggestion on improving the accessibility of our manuscript, we have changed “lattice” to “network”

-Figure 4: delete the scale on the left, it is already reported on the right side

-Thank you for your great suggestion on improving the accessibility of our manuscript, we have deleted it.

-Figure 5 (a), columns A and E: state that are details of the images of columns B and D, respectively

-Thank you for your great suggestion on improving the accessibility of our manuscript, we have added the state.

-line 368: “can be used” instead of “could available”

-Thank you for your great suggestion on improving the accessibility of our manuscript, we have changed “could available” to “can be used”

-line 369: delete “adsorption condition”

-Thank you for your great suggestion on improving the accessibility of our manuscript, we have deleted it.

-lines 370-372: better “The results showed that the S/O/W prepared with GEL (2 wt%) and without NaCas showed lamellar structure of the W phase where particles of different size of the O phase were embedded”

-Thank you for your great suggestion on improving the accessibility of our manuscript, we have redescribed the sentence: The S/O/W emulsions prepared with GEL (2 wt%) and without NaCas showed lamellar structure of the W phase where particles of different size of the O phase were embedded.

-line 379: “and” instead of “while”

-Thank you for your great suggestion on improving the accessibility of our manuscript, we have changed “while” to “and”

-line 381: delete “adsorption”

-Thank you for your great suggestion on improving the accessibility of our manuscript, we have deleted it.

-line 383: better “forming a core-shell structure, which leads to the improvement”

-Thank you for your great suggestion on improving the accessibility of our manuscript, we have redescribed the sentence: At the same time, the cross-section of microspheres showed that the presence of NaCas enhanced the thickness of the adsorption layer and forming a core-shell structure, which leads to the improvement of the physical stability of the system [44] and the re-sults were consistent with those in Fig. 5(a).

-line 448: better “A slope value for the E-π curve greater than one indicates”, and analogously at line 456

-Thank you for your great suggestion on improving the accessibility of our manuscript, we have redescribed the sentence: A slope value for the E-π curve greater than 1 indicates that the adsorption of components on the interface is non-ideal, and there is strong intermolecular interaction.

-line 455: better “The slope values for the E-π curves, reported in Figure 6c, are all “

-Thank you for your great suggestion on improving the accessibility of our manuscript, we have redescribed the sentence: The slope values for the E-π curves reported in Fig. 6(c) are all higher than 1.0.

-line 485: add the relevant reference, 54?

-Thank you for your great suggestion on improving the accessibility of our manuscript, we have added the relevant reference, 54.

-line 491: “2+” as upperscript

-感谢您对改进我们手稿的可访问性提出的重要建议,我们已经更正了它。

-第 509 行:“更密集”而不是“更密集”

-感谢您对提高手稿的可访问性提出的重要建议,我们已将“更密集”更改为“更密集”

-第510行:删除:一定厚度”

-感谢您对改善我们手稿的可访问性提出的重要建议,我们已将其删除。

-第 513 行:“乳液”比“乳液微球”更好

-感谢您对提高我们手稿的可访问性的重要建议,我们已将“乳液微球”更改为“乳液”

Reviewer 2 Report

This research is original. The paper is well organized well written and i really enjoy the reading. I try to find some point to review. Here are some of them:

1. Don't use abbreviations in the title.

2. Underline the innovative of this work in the introduction.

3. Figure 2 put the explanation of column bars inside the graph region 

I am looking forward to receive  the revised version.

Best wishes!

Author Response

Comments:

This research is original. The paper is well organized well written and i really enjoy the reading. I try to find some point to review. Here are some of them:

Comments 1:

-Don't use abbreviations in the title.

Response 1:

-Thank you for your great suggestion on improving the accessibility of our manuscript, we have changed the title to “The improvement of dispersion stability and bioaccessibility of calcium carbonate by solid/oil/water (S/O/W) emulsion.”

Comments 2:

-Underline the innovative of this work in the introduction.

Response 2:

-Thank you for your great suggestion on improving the accessibility of our manuscript, we have underlined the innovative of this work in the introduction:

Few studies on the application of traditional CaCO3 powder in liquid food have been reported. In this study, S/O/W emulsion is used to deliver CaCO3 to improve the dispersion stability and bioaccessibility. The S/O/W emulsion was stabilized by GEL and NaCas-GEL as the W phase to load CaCO3. The droplet size, microstructure, rheo-logical behavior, and interface behavior were systematically evaluated. This study aimed to investigate the stabilization mechanism of S/O/W emulsions stabilized by GEL and NaCas and to provide a theoretical basis for the delivery of CaCO3 by S/O/W emulsion and enriched the theoretical support for the development of a new nutrient delivery system.

Comments 3:

-Figure 2 put the explanation of column bars inside the graph region.

Response 3:

-Thank you for your great suggestion on improving the accessibility of our manuscript, we have put the explanation of column bars inside the graph region.

Reviewer 3 Report

The authors developed and characterized S/O/W emulsion for CaCO3 delivery. The topic is important for many readers.  Before publication some points should be clarified and improved:

1. Section 3.2. and 3.3 - numbering of figures is wrong.

2. P.8 , line 288 and 307 - the formation of NaCas-Gel binary complex has to be clarified. The sentences contradict each other. 

3. Section 3.7 and Figure 8. Statistical analysis has to be done to verify the conclusion about the improvement of bioaccessibility.

4. References - please check [J] - what is it?

Reviewer 4 Report

This study investigated the preparation of solid/oil/water (S/O/W) emulsion microspheres for the encapsulation and delivery of calcium carbonate (CaCO3) to improve the dispersion stability and bio accessibility of CaCO3. As the authors claim, the present design of S/O/W emulsion microspheres has the potential to overcome the disadvantage of poor solubility of CaCO3 and will be applied to the delivery of liquid food minerals. Therefore, I recommend the paper for minor revisions.

1.       Visual observation of the emulsions should be depicted.

2.       Evaluation of the polydispersity index (PDI) is mentioned in subsection 2.3, while its discussion is not appeared in section 3.

3. English throughout the text should be checked and revised.

Author Response

Comments:

This study investigated the preparation of solid/oil/water (S/O/W) emulsion microspheres for the encapsulation and delivery of calcium carbonate (CaCO3) to improve the dispersion stability and bio accessibility of CaCO3. As the authors claim, the present design of S/O/W emulsion microspheres has the potential to overcome the disadvantage of poor solubility of CaCO3 and will be applied to the delivery of liquid food minerals. Therefore, I recommend the paper for minor revisions.

Comments 1:

-Visual observation of the emulsions should be depicted.

Response 1:

-Thank you for your great suggestion on improving the accessibility of our manuscript, we have added the depiction of visual observation:

The S/O/W emulsions prepared by XG and NaCas-GEL binary composite were all milky white under visual observation and showed no signs of separation or foaming, and it is difficult to distinguish the difference. Therefore, the microstructure of the S/O/W emulsions prepared by NaCas and NaCas-GEL should be further observed in order to study its embedding effect and interface adsorption more directly.

Comments 2:

-Evaluation of the polydispersity index (PDI) is mentioned in subsection 2.3, while its discussion is not appeared in section 3.

Response 2:

-Thank you for your great suggestion on improving the accessibility of our manuscript, we revised subsection 2.3: Particle size measurements were performed using a laser diffractometer (S3500 Microtrac Inc, Largo USA) with the particle refractive index of 1.51 and the dispersive medium refractive index of 1.33.

Comments 3:

-English throughout the text should be checked and revised.

Response 3:

-Thank you for your great suggestion on improving the accessibility of our manuscript, we have checked and revised the text.
